# Defending Pre-trained Language Models as Few-shot Learners against Backdoor Attacks

**Zhaohan Xi**[1*]    **Tianyu Du**[2*]    **Changjiang Li**[1,3]    **Ren Pang**[1]    **Shouling Ji**[2]
**Jinghui Chen**[1]    **Fenglong Ma**[1]    **Ting Wang**[1,3]
[1]Pennsylvania State University  [2]Zhejiang University  [3]Stony Brook University
{zhaohan.xi, rbp5354, jzc5917, fenglong}@psu.edu
{zjradty, sji}@zju.edu.cn
{changjli, twang}@cs.stonybrook.edu

## Abstract

Pre-trained language models (PLMs) have demonstrated remarkable performance as few-shot learners. However, their security risks under such settings are largely unexplored. In this work, we conduct a pilot study showing that PLMs as few-shot learners are highly vulnerable to backdoor attacks while existing defenses are inadequate due to the unique challenges of few-shot scenarios. To address such challenges, we advocate MDP, a novel lightweight, pluggable, and effective defense for PLMs as few-shot learners. Specifically, MDP leverages the gap between the masking-sensitivity of poisoned and clean samples: with reference to the limited few-shot data as distributional anchors, it compares the representations of given samples under varying masking and identifies poisoned samples as ones with significant variations. We show analytically that MDP creates an interesting dilemma for the attacker to choose between attack effectiveness and detection evasiveness. The empirical evaluation using benchmark datasets and representative attacks validates the efficacy of MDP. Code available at `https://github.com/z haohan-xi/PLM-prompt-defense`.

## 1 Introduction

The prompt-based learning paradigm is revolutionizing the ways of using pre-trained language models (PLMs)[7, 27, 28, 1] in various NLP tasks. Unlike the conventional fine-tuning paradigm that requires re-training the PLM, the prompt-based paradigm reformulates the downstream task as a masked language modeling problem and uses proper prompts to coax the model to produce textual outputs[17]. For example, to analyze the sentiment of a movie review, one may append the prompt "the movie is ___" to the given review and guide the model to predict the missing sentiment word (e.g., "terrible" or "great"). Recent work shows that with proper prompting, even moderate-sized PLMs can be adapted as performant few-shot learners when training data is limited[9].

In contrast to its increasing popularity, the security implications of this prompt-based paradigm are largely under-explored. Recent work[8, 34, 2] shows that similar to their fine-tuned counterparts, prompt-based PLMs are susceptible to textual backdoor attacks, in which misclassification rules are injected into PLMs, only to be activated by poisoned samples containing "triggers" (e.g., the rare word of "cr"). However, how to effectively mitigate such threats, especially under the few-shot setting, remains an open challenge.

In this work, we conduct a pilot study showing that few-shot scenarios entail unique challenges for defending against textual backdoor attacks, including scarce training data, intricate interactions with prompts, and limited computational capacity. For instance, many existing defenses[3, 24, 36] designed for the fine-tuning paradigm require reliable statistical estimates of the downstream datasets

---

*Equal contribution.

37th Conference on Neural Information Processing Systems (NeurIPS 2023).

and therefore perform poorly under the few-shot setting. Thus, it necessitates developing effective defenses tailored to the setting of few-shot learning.

Towards this end, we advocate MDP (masking-differential prompting), an effective, lightweight, and pluggable backdoor defense for PLMs as few-shot learners. At a high-level, MDP leverages the key observation that compared with clean samples, poisoned samples often show higher sensitivity to random masking: if its trigger is (partially) masked, the language modeling probability of a poisoned sample tends to vary greatly. Therefore, with reference to the limited few-shot data as "distributional anchors", MDP compares the representations of given samples under varying masking and identifies poisoned samples as ones with significant variations. To boost its effectiveness, MDP (optionally) optimizes the prompt to further improve the masking-invariance of clean samples.

To validate its effectiveness, we empirically evaluate MDP using benchmark datasets and representative attacks. The results show that MDP effectively defends PLMs against various attacks under the few-shot setting, with little impact on their performance in downstream tasks. Moreover, we show analytically that MDP creates an interesting dilemma for the attacker to choose between attack effectiveness and detection evasiveness.

To summarize, this work makes the following contributions.

- To our best knowledge, this is the first work on defending PLMs as few-shot learners against backdoor attacks. We reveal that the few-shot setting entails unique challenges while existing defenses for the fine-tuning paradigm are not easily retrofitted to its specificities.

- We propose MDP, a novel defense tailored to the few-shot setting. Leveraging the gap between the masking sensitivity of clean and poisoned samples and utilizing the few-shot data to effectively estimate such sensitivity, MDP detects poisoned samples with high accuracy at inference time.

- Using benchmark datasets and representative attacks, we empirically validate that MDP outperforms baseline defenses by large margins while causing little impact on the performance of LMs in downstream tasks.

## 2   Related Work

We survey the literature relevant to this work in the categories of few-shot learning, PLM prompting, and textual backdoor attacks and defenses.

**Few-shot learning** [32] enables pre-trained models to generalize to new tasks using only a few (labeled) samples. In the NLP domain, typical few-shot learning methods include meta-learning [40], intermediate training [38, 39], and semi-supervised learning [20, 33]. Recently, prompt-based learning [23] receives increasing attention since the introduction of GPT-3 [1], which demonstrates remarkable few-shot performance by using natural-language prompts and task demonstrations to contextualize inputs [17, 9, 41, 13, 18].

**PLM prompting** treats downstream tasks as masked language modeling problems and leverages prompts to guide PLMs to produce textual outputs [23]. With proper prompting, even moderate-sized PLMs function as performant few-shot learners [9]. While manually designing prompts requires domain expertise and is often sub-optimal [1, 23], recent work explores generating prompts automatically [13, 18, 16, 44]. For instance, P-Tuning [17] and DART [41] define prompts as pseudo-tokens and optimize prompts in the continuous space, achieving state-of-the-art performance.

**Textual backdoor attacks** extend the attacks proposed in the computer vision domain [11, 5, 22] to NLP tasks. By polluting training data or modifying model parameters (e.g., embeddings), the attacks inject misclassification rules into language models, which are activated at inference by poisoned samples containing "triggers" such as rare words [12, 35, 42, 43, 37], natural sentences [6, 4], and specific patterns [26, 21]).

**Textual backdoor defenses** aim to defend LMs against backdoor attacks. For instance, based on the observation that trigger words tend to dominate poisoned samples, STRIP [10] detects poisoned samples at run-time as ones with stable predictions under perturbation. As trigger words often increase the perplexity of poisoned samples, ONION [24] identifies poisoned samples by inspecting the perplexity changes of given samples under word deletion. RAP [36] leverages the difference

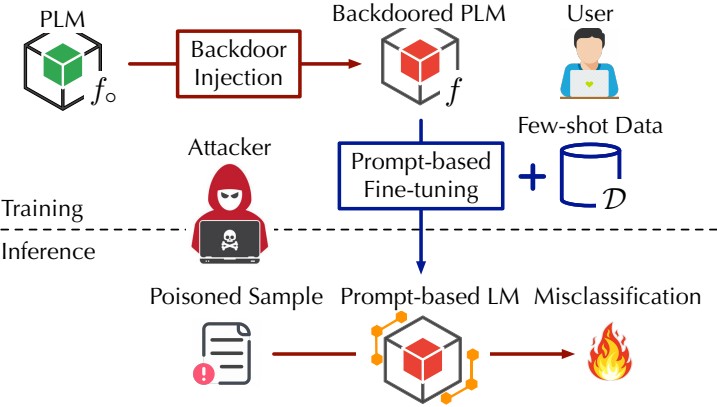

Figure 1: Threat model of NLP backdoor attacks: the attacker injects a backdoor into the PLM $f$; the victim user adapts $f$ as a few-shot learner in the downstream task; the attacker activates the backdoor via feeding $f$ with poisoned samples.

between the robustness of clean and poisoned samples to crafted perturbation and injects extra triggers into given samples to detect poisoned samples.

However, most existing defenses are designed for the fine-tuning paradigm. How to mitigate the threat of textual backdoor attacks for the prompt-based paradigm, especially under the few-shot setting, remains an open challenge. This work represents a solid initial step to bridge this gap.

## 3   Background

We present the key concepts and assumptions used throughout the paper.

### 3.1   Few-shot Prompting

Let $X_{\text{in}} = \{x_1, x_2, \ldots, x_n\}$ be an input sample, in which $x_i$ is the $i$-th token and $n$ is the length of $X_{\text{in}}$. In prompt-based learning, $X_{\text{in}}$ is padded with a template $\mathcal{T}$ to form a prompt:

$$X_{\text{prompt}} = [\texttt{cls}]\, X_{\text{in}}\, [\texttt{sep}]\, \mathcal{T}\, [\texttt{sep}] \tag{1}$$

where $\mathcal{T}$ is a task-specific string template containing a masked token:

$$\mathcal{T} = [T_{1:i}]\, [\texttt{mask}]\, [T_{i+1:m}] \tag{2}$$

The existing methods differ in the definition of the template $\mathcal{T}$. In discrete prompts [23], $[T_i]$ are selected from the vocabulary $\mathcal{V}$, while in continuous prompts [18], $[T_i]$ are defined as pseudo tokens.

Given $X_{\text{prompt}}$, the PLM $f$ (parameterized by $\theta$) is guided to output the token distribution of the masked token $p_\theta([\texttt{mask}]|X_{\text{prompt}})$. The probability that $X_{\text{in}}$ belongs to a class $y \in \mathcal{Y}$ is predicted as:

$$p_\theta(y|X_{\text{prompt}}) = \sum_{v \in \mathcal{V}_y} p_\theta([\texttt{mask}] = v|X_{\text{prompt}}) \tag{3}$$

where $\mathcal{V}_y$ is the set of label tokens related to $y$.

Under the few-shot setting, the user has access to a limited training set (e.g., $K = 16$ samples per class) and searches for the template $\mathcal{T}$ that optimizes the accuracy of $f$ in the downstream task (yet without modifying $\theta$).

### 3.2   Threat Model

As illustrated in Figure 1, we consider a malicious model provider as the attacker, who injects a backdoor into the PLM $f_\circ$ and releases the backdoored model $f$. We focus on the targeted-attack case in which the backdoor is defined as classifying samples with triggers ("poisoned samples") to a target class $t$ desired by the attacker. The victim user downloads $f$ and applies it as a prompt-based few-shot learner in the downstream task. The attacker activates the backdoor at inference time by

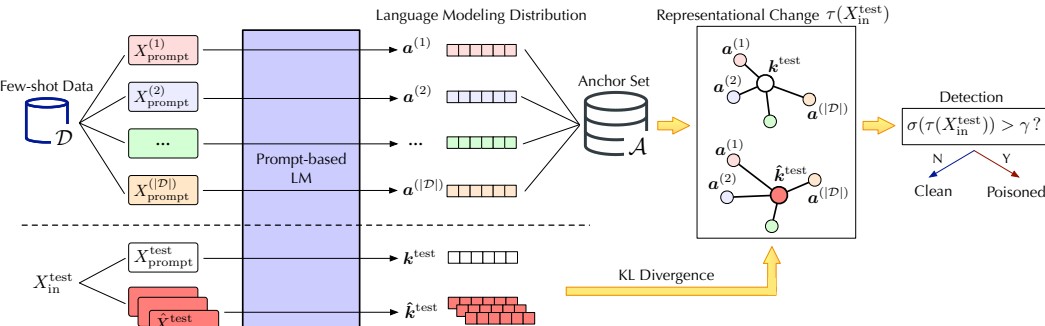

Figure 2: Overview of MDP: it detects a given sample $X_{\text{in}}^{\text{test}}$ as poisoned or clean by measuring the variation of its representational change with respect to a set of distributional anchors $\mathcal{A}$.

feeding $f$ with poisoned samples. To simulate the worst-case scenario for the defenses, we assume the attacker has access to the downstream dataset and injects the backdoor into the PLM using a fine-tuning approach. Formally, the attack is formulated as the following optimization objective:

$$\min_{\theta} \mathbb{E}_{(x,y) \in \mathcal{D}_{\text{c}}} \ell(f_\theta(x), y) + \lambda \mathbb{E}_{(\tilde{x}, t) \in \mathcal{D}_{\text{p}}} \ell(f_\theta(\tilde{x}), t) \tag{4}$$

where $\mathcal{D}_{\text{c}}$ and $\mathcal{D}_{\text{p}}$ respectively refer to the clean and poisoning data and $\ell$ is the loss function (e.g., cross-entropy). Intuitively, the first term ensures $f$ functions normally on clean samples, the second term ensures $f$ classifies poisoned samples to the target class $t$, and $\lambda$ is a hyper-parameter to balance the two objectives.

Compared with prior work [10, 24, 36], we consider a more realistic and challenging setting: as the defender, the victim user only has limited few-shot data and computational capacity. Further, the user has no knowledge about the attacker's training procedure, attack strategy, or trigger definition.

## 4  MDP

Next, we present MDP, a novel backdoor defense for PLMs as few-shot learners.

### 4.1  Overview of MDP

At a high level, MDP exploits the observation that compared with clean samples, poisoned samples often show higher sensitivity to random masking (i.e., randomly selecting and substituting a token with [mask]). Intuitively, by the design of backdoor attacks, the trigger dominates a poisoned sample and forces it to be classified to the target class. Thus, if the trigger is (partially) masked, the language modeling probability of a poisoned sample tends to vary greatly. In comparison, a clean sample is often less sensitive to random masking. It is therefore feasible to distinguish clean and poisoned samples by comparing their masking sensitivity.

A naïve approach to measure the masking sensitivity is to compare the model prediction (i.e., "positive" and "negative") of a given sample before and after masking, which however fails to capture the complex variation of the language modeling probability (details in §5.4). Instead, MDP uses the limited few-shot data as "distributional anchors" and measures the representational change of the sample under varying masking, as illustrated in Figure 2. To further boost its distinguishing power, MDP optimizes the prompt to improve the masking-invariance of clean samples. Below we elaborate on the design and implementation of MDP, with the complete algorithm deferred to §A.

### 4.2  Modeling Masking Sensitivity

To quantify the representational change of a given sample under masking, we leverage the limited few-shot data $\{(X_{\text{in}}^{(i)}, y^{(i)})\}$ as a set of "distributional anchors". Specifically, for each $X_{\text{in}}^{(i)}$, we generate its prompt $X_{\text{prompt}}^{(i)}$ to query the PLM and obtain the distribution as in Eq. 3:

$$\boldsymbol{a}^{(i)} = p_\theta(v | X_{\text{prompt}}^{(i)}) \quad (v \in \mathcal{V}) \tag{5}$$

Note that rather than mapping it back to the label space $\mathcal{Y}$, we cache the entire language modeling distribution as the representation of $X_{\text{in}}^{(i)}$ and consider the data store $\mathcal{A} = \{\boldsymbol{a}^{(i)}\}$ as the anchor set.

At run-time, for a given sample $X_{\text{in}}^{\text{test}}$, we construct its prompt $X_{\text{prompt}}^{\text{test}}$ and query the model to obtain its distribution $\boldsymbol{k}^{\text{test}} = p_\theta(v|X_{\text{prompt}}^{\text{test}})$. We measure the distance between $X_{\text{in}}^{\text{test}}$ and the anchors by the Kullback–Leibler divergence between $\boldsymbol{k}^{\text{test}}$ and each $\boldsymbol{a}^{(i)}$: $D_{\text{KL}}(\boldsymbol{k}^{\text{test}}\|\boldsymbol{a}^{(i)})$. We regard the vector $\boldsymbol{d}(X_{\text{in}}^{\text{test}}) = [D_{\text{KL}}(\boldsymbol{k}^{\text{test}}\|\boldsymbol{a}^{(i)})]$ as the coordinates of $X_{\text{in}}^{\text{test}}$ with respect to the anchors.

Let $\hat{X}_{\text{in}}^{\text{test}}$ be the masked version of $X_{\text{in}}^{\text{test}}$ under random masking. Following the procedure above, we compute the coordinates of $\hat{X}_{\text{in}}^{\text{test}}$ as $\boldsymbol{d}(\hat{X}_{\text{in}}^{\text{test}})$. We measure the representational change due to masking by the difference of $\boldsymbol{d}(\hat{X}_{\text{in}}^{\text{test}})$ and $\boldsymbol{d}(X_{\text{in}}^{\text{test}})$:

$$\tau(X_{\text{in}}^{\text{test}}) = \Delta(\boldsymbol{d}(\hat{X}_{\text{in}}^{\text{test}}), \boldsymbol{d}(X_{\text{in}}^{\text{test}})) \tag{6}$$

Empirically, we find the Kendall rank coefficient as an effective similarity function $\Delta$, which measures the rank correlation between $\boldsymbol{d}(X_{\text{in}}^{\text{test}})$ and $\boldsymbol{d}(\hat{X}_{\text{in}}^{\text{test}})$ (i.e., the relative proximity between $X_{\text{in}}^{\text{test}}$ and different anchors) and is insensitive to concrete KL-divergence measures.

We then measure the variation of $\tau(X_{\text{in}}^{\text{test}})$ under varying masking to quantify the masking sensitivity of $X_{\text{in}}^{\text{test}}$ and detect it as a poisoned sample if its variation is above a pre-defined threshold $\gamma$.

### 4.3 Amplifying Masking Invariance

Recall that MDP distinguishes clean and poisoned samples based on the gap between their sensitivity to random masking. To further boost its distinguishing power, we (optionally) optimize the prompt to improve the masking invariance of clean samples.

Specifically, given few-shot data $\{(X_{\text{in}}, y)\}$, let $\hat{X}_{\text{in}}$ be the masked version of $X_{\text{in}}$ and $\hat{X}_{\text{prompt}}$ and $X_{\text{prompt}}$ be their prompts. We define the masking-invariant constraint as:

$$\mathcal{L}_{\text{MI}} = \mathbb{E}_{X_{\text{in}}, \text{mask}(\cdot)} \ell(f_\theta(\hat{X}_{\text{prompt}}), f_\theta(X_{\text{prompt}})) \tag{7}$$

where the expectation is taken over the few-shot data $X_{\text{in}}$ and random masking $\text{mask}(\cdot)$. Intuitively, $\mathcal{L}_{\text{MI}}$ encourages the model to generate similar distributions for a clean sample under varying masking. Note that $\mathcal{L}_{\text{MI}}$ is pluggable into any prompt-based learning methods including P-Tuning [17] and DART [41] to complement other optimization objectives.

### 4.4 Theoretical Justification

Next, we provide theoretical justification for the effectiveness of MDP. To simplify the analysis, we assume the following setting: given a binary classification task and a vocabulary of two tokens {+, -}, a sample $X_{\text{in}}$ is classified as 1 if $p_\theta(+|X_{\text{in}}) > \frac{1}{2}$ and 0 otherwise; a poisoned sample $X_{\text{in}}$ (with target class $t = 1$) comprises $n$ tokens (including one trigger token); in its masked variant $\hat{X}_{\text{in}}$, one token is randomly masked; a single anchor $X_{\text{in}}^*$ is used as the reference, with $p^* \triangleq p_\theta(+|X_{\text{in}}^*)$. Theorem 4.1 reveals that there exists a trade-off between attack effectiveness and detection evasiveness (proof deferred to §B).

**Theorem 4.1.** *Assume i) the attack is effective – if a non-trigger token is masked, $p_\theta(+|\hat{X}_{\text{in}}) \geq \kappa^+ > \frac{1}{2}$, and ii) a clean sample is masking-invariant – if the trigger token is masked, $p_\theta(+|\hat{X}_{\text{in}}) \leq \kappa^- < \frac{1}{2}$, and if the detection threshold $\gamma$ is set on the variation of the representational change of $X_{\text{in}}$ under random masking, then to evade the detection, it satisfies:*

$$|h(\kappa^+) - h(\kappa^-)| \leq \frac{n}{\sqrt{n-1}}\gamma \tag{8}$$

*where $h(\cdot)$ is defined as the KL divergence function with respect to $p^*$:*

$$h(p) \triangleq p \log \frac{p}{p^*} + (1-p) \log \frac{1-p}{1-p^*} \tag{9}$$

Intuitively, for the attack to be effective, $\kappa^+$ should be large; however, to evade the detection, $\kappa^+$ is upper-bounded by Eq. 8. Thus, MDP creates an interesting dilemma for the attacker to choose between attack effectiveness and detection evasiveness. Moreover, if the model is both accurate in classifying clean samples (i.e., $\kappa^-$ is sufficiently small) and masking-invariant with respect to clean

samples (i.e., $\gamma$ can be set sufficiently small without incurring false positive cases), which makes the following condition hold:

$$|h(\kappa^-) + 1 + \frac{1}{2}\log p^*(1-p^*)| > \frac{n}{\sqrt{n-1}}\gamma, \tag{10}$$

it is then impossible to launch effective attacks without being detected because $\kappa^+$ can not satisfy the two objectives simultaneously (proof in §B).

## 5 Empirical Evaluation

### 5.1 Experimental Setting

**Datasets.** We conduct the evaluation across 5 sentence classification datasets (SST-2, MR, CR, SUBJ, TREC) widely used to benchmark prompt-based few-shot learning methods [9, 17, 41]. We follow the same setting of LM-BFF [9], which samples $K = 16$ samples per class to form the training and validation sets respectively. The dataset statistics are summarized in Table 1.

| Dataset | # Classes | Avg. Len | Train | Dev | Test |
|---------|-----------|----------|-------|-----|------|
| SST-2 | 2 | 15.6 words | 6.9k | 0.9k | 1.8k |
| MR | 2 | 21.0 words | 8.0k | 0.7k | 2.0k |
| CR | 2 | 20.1 words | 1.5k | 0.3k | 2.0k |
| SUBJ | 2 | 24.1 words | 7.0k | 1.0k | 2.0k |
| TREC | 6 | 10.0 words | 5.0k | 0.5k | 0.5k |

Table 1. Statistics of the datasets used in the experiments.

**Models.** A victim model comprises a PLM and a prompt model. We use RoBERTa-large [19] as the PLM, which is widely used in prompt-based learning [9, 29, 41, 44], and DART [41] as the prompt model, which achieves state-of-the-art performance under the few-shot setting.

**Attacks.** We use 5 representative textual backdoor attacks to evaluate MDP and other defenses.

BadNets [11] is originally designed as a backdoor attack in the computer vision domain and extended to NLP tasks by selecting rare words as triggers [12]. AddSent [6] is similar to BadNets but uses neutral sentences as triggers to make poisoned samples stealthier. EP [35] perturbs the embeddings of trigger words rather than modifying the PLM parameters. LWP [15] uses a layer-wise weight poisoning strategy to only poison the first layers of PLMs with combinatorial triggers. SOS [37] defines the triggers as the co-occurrence of multiple pre-defined words, which are further inserted into natural sentences to make the attacks more evasive.

**Baseline defenses.** As MDP represents the first backdoor defense for the prompt-based paradigm, we adapt 3 representative defenses designed for the fine-tuning paradigm as the baselines.

Based on the observation that the prediction of a poisoned sample is often dominated by the trigger, STRIP [10] detects poisoned samples as ones with stable predictions under perturbation. ONION [24] relies on the hypothesis that the trigger is out of the context of a poisoned sample, and detects poisoned samples by inspecting the perplexity change under word deletion. RAP [36] leverages the gap between the robustness of clean and poisoned samples to perturbation and injects crafted perturbation into given samples to detect poisoned samples. The detailed description of the baselines is deferred to §C.

### 5.2 Implementation Details

To simulate a challenging scenario, we assume the attacker has access to the full training sets (cf. Table 1) and injects backdoors into PLMs by fine-tuning the models. The attack setting (e.g., trigger definitions) is summarized in §C. We apply MDP and baselines on the backdoored PLMs under the few-shot, prompt-based learning paradigm; that is, the defender has only access to the few-shot data ($K = 16$ samples per class). We apply a grid search over the hyperparameters to select the optimal setting for each defense.

Following previous studies [10, 36], the attack performance is evaluated using the metrics of i) clean accuracy (CA), defined as the victim model's accuracy on clean samples, and ii) attack success rate (ASR), defined as its accuracy of classifying poisoned samples to the target label desired by the

attacker. Intuitively, CA and ASR respectively quantify the model's performance on the original and backdoor tasks. Meanwhile, the defense performance is evaluated by the metrics of i) false rejection rate (FRR), defined as the percentage of clean samples that are mistakenly labeled as poisoned, ii) false acceptance rate (FAR), defined as the percentage of poisoned samples that are mislabeled as clean, and iii) the area under the ROC curve (AUC), an aggregate measure of performance across all possible classification thresholds. All the measures are averaged across five sampled training sets as in LM-BFF [9].

| Dataset | Attack | CA (%) | ASR (%) | STRIP | | ONION | | RAP | | MDP | |
|---|---|---|---|---|---|---|---|---|---|---|---|
| | | | | FRR | FAR | FRR | FAR | FRR | FAR | FRR | FAR |
| SST-2 | BadNets | 95.06 | 94.38 | 7.56 | 87.44 | 2.78 | 9.28 | 3.11 | 64.28 | 5.33 | 1.77 |
| | AddSent | 94.45 | 100.0 | 2.75 | 72.56 | 7.06 | 26.72 | 5.61 | 37.50 | 4.45 | 3.53 |
| | LWP | 93.41 | 95.53 | 5.96 | 89.39 | 8.28 | 7.39 | 0.83 | 43.77 | 5.27 | 4.78 |
| | EP | 93.63 | 95.95 | 1.72 | 72.06 | 5.28 | 12.89 | 2.72 | 58.11 | 5.05 | 0.73 |
| | SOS | 91.65 | 92.41 | 2.98 | 87.56 | 4.06 | 32.56 | 1.89 | 51.28 | 0.00 | 0.00 |
| MR | BadNets | 89.80 | 98.30 | 11.70 | 72.30 | 4.80 | 15.60 | 2.75 | 25.35 | 5.10 | 5.60 |
| | AddSent | 89.60 | 97.50 | 16.20 | 60.00 | 4.65 | 37.25 | 9.35 | 39.70 | 5.05 | 10.90 |
| | LWP | 89.65 | 96.90 | 9.35 | 82.70 | 1.60 | 17.45 | 1.70 | 52.55 | 5.25 | 3.60 |
| | EP | 89.40 | 96.60 | 2.20 | 88.90 | 15.35 | 12.60 | 6.45 | 70.60 | 4.70 | 3.00 |
| | SOS | 89.85 | 97.30 | 5.20 | 75.90 | 0.90 | 64.10 | 15.20 | 58.85 | 4.85 | 3.40 |
| CR | BadNets | 89.95 | 92.30 | 2.85 | 98.70 | 5.20 | 7.45 | 1.35 | 43.60 | 4.95 | 5.10 |
| | AddSent | 91.45 | 95.70 | 10.10 | 62.20 | 4.75 | 19.50 | 12.95 | 48.90 | 4.80 | 3.00 |
| | LWP | 89.75 | 91.30 | 1.80 | 99.10 | 4.90 | 27.85 | 4.05 | 39.20 | 5.10 | 3.50 |
| | EP | 89.35 | 97.55 | 2.20 | 87.20 | 10.15 | 4.40 | 7.65 | 45.20 | 5.35 | 9.40 |
| | SOS | 91.45 | 100.0 | 2.20 | 78.20 | 0.75 | 37.55 | 3.40 | 55.30 | 0.20 | 0.00 |
| SUBJ | BadNets | 96.05 | 94.20 | 5.10 | 68.85 | 3.50 | 16.60 | 12.40 | 43.65 | 5.30 | 7.90 |
| | AddSent | 95.90 | 97.00 | 2.50 | 85.50 | 4.30 | 34.20 | 7.30 | 68.20 | 4.85 | 9.00 |
| | LWP | 96.15 | 99.10 | 4.55 | 98.70 | 4.65 | 7.40 | 1.00 | 18.60 | 5.40 | 10.90 |
| | EP | 96.70 | 99.90 | 4.75 | 99.10 | 5.25 | 4.10 | 4.70 | 33.25 | 4.90 | 10.30 |
| | SOS | 94.90 | 99.60 | 5.15 | 75.50 | 4.90 | 61.30 | 0.10 | 29.10 | 5.35 | 4.10 |
| TREC | BadNets | 93.00 | 95.30 | 4.30 | 73.76 | 5.40 | 54.53 | 5.55 | 50.61 | 4.80 | 2.49 |
| | AddSent | 96.60 | 93.65 | 5.20 | 79.28 | 4.80 | 36.74 | 3.55 | 47.60 | 3.60 | 7.18 |
| | LWP | 94.40 | 97.24 | 5.60 | 99.17 | 4.60 | 25.69 | 1.23 | 93.09 | 5.20 | 4.42 |
| | EP | 95.80 | 97.51 | 4.60 | 63.81 | 5.20 | 11.22 | 10.43 | 42.68 | 4.80 | 5.25 |
| | SOS | 91.80 | 99.45 | 5.20 | 68.78 | 4.40 | 80.61 | 14.83 | 63.71 | 4.60 | 4.97 |

Table 2. Performance of MDP and baseline defesens on 5 datasets, with lower FAR/FRR indicating better defense performance. The detection threshold is set based on the allowance of 5% FRR on the training set.

## 5.3 Main Results

We first evaluate the effectiveness of various backdoor attacks under prompt-based fine-tuning, with results summarized in Table 2. Observe that across all the datasets, most attacks attain both CA and ASR above 90%, indicating their effectiveness in the downstream and backdoor tasks.

We then compare the performance of MDP and baselines in defending against these attacks. For each defense, we set the detection threshold (e.g., the variation threshold for MDP) based on the allowance of 5% FRR on the training set, and report its FAR and FRR on the testing set. In the case of ONION, following prior work [36], we evaluate different thresholds of perplexity change and select the threshold that approximately achieves 5% FRR on the training set.

Table 2 summarizes the main results (additional results in §D). Observe that MDP attains the lowest FARs against all the attacks across all the datasets and outperforms baselines by large margins. In particular, it achieves near-perfect defenses against the SOS attack on the SST-2 and CR datasets. The observation confirms the effectiveness of MDP in detecting poisoned samples, which is mainly attributed to that i) the clean and poisoned samples show discernible sensitivity to random masking and ii) MDP effectively utilizes the few-shot data as anchors to measure such sensitivity.

In comparison, the baseline defenses are less effective, with FARs over 90% in many cases. This may be explained by the conflict between the limited few-shot data and the reliance of these defenses on sufficient training data. Specifically, to measure the prediction stability of a given sample under perturbation, STRIP randomly replaces a fraction of its words with ones from a training sample

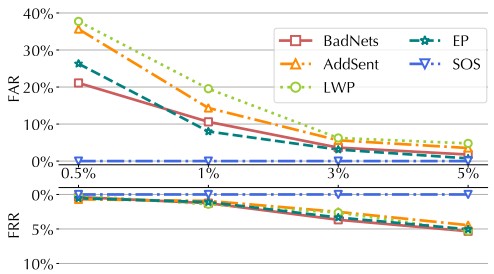

Figure 3: Performance of MDP on SST-2 with different FRR allowances on the training set; baseline defenses all have FARs above 50% (not shown).

| Dataset | | Attack | | | | |
|---|---|---|---|---|---|---|
| | | BadNets | AddSent | LWP | EP | SOS |
| SST-2 | FRR | 5.07 | 5.29 | 5.39 | 5.39 | 5.17 |
| | FAR | 24.89 | 58.37 | 55.50 | 47.82 | 73.28 |
| MR | FRR | 5.40 | 5.05 | 5.45 | 5.15 | 4.60 |
| | FAR | 72.80 | 74.80 | 55.00 | 52.10 | 80.80 |
| CR | FRR | 4.40 | 5.10 | 4.80 | 5.45 | 5.25 |
| | FAR | 83.10 | 75.30 | 73.80 | 52.20 | 56.10 |
| SUBJ | FRR | 5.40 | 4.25 | 4.60 | 4.75 | 5.35 |
| | FAR | 9.80 | 67.40 | 14.90 | 15.70 | 37.90 |
| TREC | FRR | 5.20 | 4.90 | 4.90 | 4.70 | 5.20 |
| | FAR | 75.14 | 71.55 | 43.37 | 70.44 | 26.52 |

Table 3: Performance of MDP using prediction variance as the masking-sensitivity measure.

that have the highest frequency-inverse document frequency (TF-IDF) scores. However, due to the limited number of training samples, both the substitution words and the estimated TF-IDF scores tend to be highly biased, which negatively impacts the performance of STRIP. ONION removes outlier words that cause sharp perplexity changes before inference, which is inherently ineffective against complex triggers (e.g., natural sentences) [6]. Moreover, the threshold for detecting outlier words can be significantly biased by the limited training samples under the few-shot setting. RAP trains a word-based robustness-aware trigger such that inserting this trigger causes significant prediction changes for clean samples but not for poisoned samples. However, under the few-shot setting, the optimality of the RAP trigger is largely limited by the available few-shot data, which negatively affects its detection effectiveness.

## 5.4 Influential Factors

We conduct additional studies to understand the impact of key factors on MDP. Due to space limitations, we mainly present the results on SST-2, with other results deferred to §D.

**FRR allowance.** We adjust the detection threshold corresponding to varying FRR allowance on the training set. Figure 3 shows that MDP maintains its superior performance under different FRRs (0.5%, 1%, and 3%). In comparison, the baselines all have FARs above 50% (not shown).

**Sensitivity measures.** Instead of using the few-shot data as distributional anchors to measure the masking sensitivity of a given sample $X_{\mathrm{in}}^{\mathrm{test}}$, we use its prediction variance due to masking as the sensitivity measure. Specifically, given the prediction of $X_{\mathrm{in}}^{\mathrm{test}}$: $y = \arg\max_{y'} p_\theta(y'|X_{\mathrm{in}}^{\mathrm{test}})$, we measure the confidence variance of the masked variant $\hat{X}_{\mathrm{in}}^{\mathrm{test}}$ with respect to $y$: $\sigma(p_\theta(y|\hat{X}_{\mathrm{in}}^{\mathrm{test}}))$. Intuitively, a poisoned sample tends to have a larger variance since masking the trigger may cause the prediction to fluctuate significantly. Following the same setting in §5.3, we set the threshold based on 5% FRR allowance on the training set and evaluate MDP on the testing set. Table 3 shows that using the alternative sensitivity measure causes the performance of MDP to drop sharply (cf. Table 2). For instance, its FAR increases by over 50% against LWP. The results confirm our analysis that simple statistics such as prediction confidence may fail to capture the complex variation of the language modeling probability due to masking.

**Masking-invariance constraints.** Recall that the masking-invariant constraint $\mathcal{L}_{\mathrm{MI}}$ is designed to improve the masking invariance of clean samples. Here, we evaluate its impact on MDP's overall performance. Specifically, we adjust the weight of $\mathcal{L}_{\mathrm{MI}}$ in the prompt optimization [41] from 0.25 to 4. For each distinct weight, we set the detection threshold based on 5% FRR allowance on the training set and report its performance on the testing set. As shown in Figure 4, as the weight of $\mathcal{L}_{\mathrm{MI}}$ varies, the FARs of MDP against most attacks first drop and then gradually increase. This observation may be explained as follows. With an overly small weight, $\mathcal{L}_{\mathrm{MI}}$ has little effect on improving the masking invariance of clean samples, while overly emphasizing $\mathcal{L}_{\mathrm{MI}}$ negatively impacts the classification accuracy, resulting in higher FARs. It is thus crucial to properly calibrate the weight of $\mathcal{L}_{\mathrm{MI}}$ to optimize the performance of MDP.

**Few-shot data size.** We further evaluate how the few-shot data size (i.e., shots) influences the performance of MDP. Besides the default shots ($K = 16$ per class) used in the previous evaluation, we vary $K$ from 4 to 64 to build the anchor set and evaluate MDP, in which the FRR allowance on

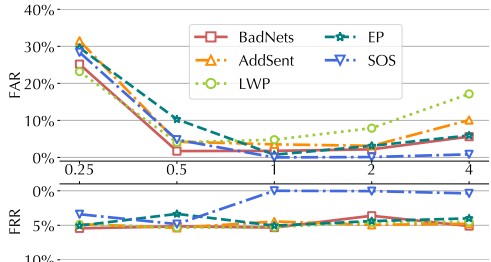
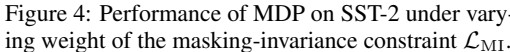
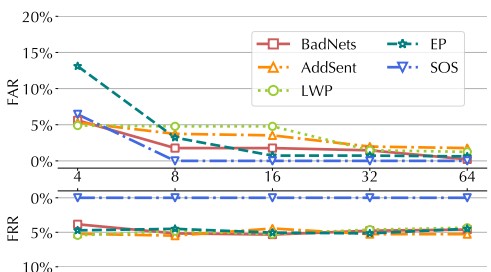

Figure 4: Performance of MDP on SST-2 under varying weight of the masking-invariance constraint $\mathcal{L}_{\text{MI}}$.

Figure 5: Performance of MDP on SST-2 with varying size of few-shot data ($K$ samples per class).

the training set is fixed as 5%. Figure 5 reports the performance of MDP under varying shots $K$. Observe that its FARs steadily improve as $K$ increases. Intuitively, with a larger anchor set, MDP quantifies the representational variation of given samples due to random masking more precisely, leading to more accurate detection. Also, notice that $K = 16$ is often sufficient for MDP to obtain satisfactory performance.

**Prompt types.** We further evaluate the impact of prompt types on MDP. Recall that in discrete prompts [23], the tokens in the prompt template are selected from the vocabulary, while in continuous prompts [18], the tokens are pseudo-tokens and optimized in a continuous space. Table 4 evaluates MDP on discrete prompt-based models. Compared with continuous prompts (cf. Table 2), MDP is less effective under discrete prompts. For instance, its FAR against BadNets on MR increases by 17%. This may be explained by that continuous prompts entail larger spaces to better optimize the masking invariance constraint, suggesting that using differentiable, continuous prompts benefits MDP in defending against backdoor attacks.

| Dataset | | Attack | | | | |
|---------|-----|---------|---------|-------|-------|-------|
| | | BadNets | AddSent | LWP | EP | SOS |
| SST-2 | FRR | 5.27 | 4.39 | 5.15 | 5.11 | 0.00 |
| | FAR | 5.09 | 19.02 | 18.40 | 10.08 | 0.00 |
| MR | FRR | 5.45 | 4.85 | 5.05 | 5.15 | 5.45 |
| | FAR | 22.60 | 32.80 | 24.20 | 14.50 | 27.80 |
| CR | FRR | 3.80 | 5.30 | 5.45 | 5.15 | 4.45 |
| | FAR | 14.40 | 33.50 | 20.10 | 24.40 | 11.00 |
| SUBJ | FRR | 5.40 | 4.75 | 5.20 | 5.00 | 5.25 |
| | FAR | 11.70 | 31.10 | 12.00 | 32.40 | 25.10 |
| TREC | FRR | 5.00 | 4.10 | 4.50 | 5.30 | 4.50 |
| | FAR | 16.02 | 37.85 | 32.60 | 23.48 | 26.80 |

Table 4. Performance of MDP on discrete prompt-based models (with 5% FRR allowance on the training set).

**PLMs.** Besides RoBERTa-large [19], we also consider alternative PLMs including BART [14] and GPT-2 [27] as the victim models. Table 5 shows MDP's performance on such alternative PLMs against BadNets and AddSent attacks (with respect to SST-2). Observe that MDP has comparable performance on BART and GPT-2 to that on RoBERTa-large, indicating that it generalizes to other PLMs as well.

| PLM | Attack | CA | ASR | FRR | FAR |
|-----|--------|-------|-------|------|-------|
| BART | BadNets | 94.03 | 98.79 | 2.85 | 11.42 |
| | AddSent | 92.07 | 99.12 | 6.82 | 16.61 |
| GPT-2 | BadNets | 92.78 | 91.28 | 5.60 | 6.24 |
| | AddSent | 92.31 | 91.28 | 4.05 | 9.46 |

Table 5. MDP on alternative PLMs.

**Masking rate.** Recall that the masking sensitivity influences MDP's performance (§4.2). Here we vary the masking rate from 0.1 to 0.4 (with 0.2 as the default). Figure 6 presents MDP's FAR and FRR on SST-2 with respect to different attacks. Notably, properly setting the masking rate is critical for MDP. This can be explained as follows. An overly small masking rate may lower the probability of "hitting" the triggers, while an overly large masking rate may significantly alter the semantics of given samples after masking, both making clean and poisoned samples less distinguishable.

### 5.5 Alternative Scenarios

Thus far we assume that (i) the backdoor is injected into the PLM, (ii) the trigger is defined as explicit perturbation (e.g., adding words or sentences), and (iii) the attacker is unaware of MDP. Next, we evaluate MDP in alternative scenarios in which these assumption do not hold, with results summarized in Table 7.

**Alternative threat models.** BadPrompt [2] injects the backdoor into the prompt and releases the end-to-end PLM to the victim user, assuming that the user directly uses the model without further

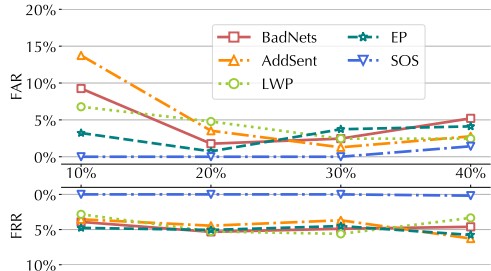

Figure 6: MDP on SST-2 with varying masking rate.

| Dataset | Attack | CA | ASR | FRR | FAR |
|---|---|---|---|---|---|
| | BadPrompt | 87.50 | 81.39 | 7.16 | 8.28 |
| SST-2 | StyleAttack | 92.97 | 78.67 | 2.94 | 24.33 |
| | Adaptive Attack | 94.04 | 82.71 | 6.94 | 13.78 |
| | BadPrompt | 84.75 | 90.10 | 4.45 | 6.70 |
| MR | StyleAttack | 88.65 | 84.10 | 2.95 | 31.30 |
| | Adaptive Attack | 89.65 | 86.45 | 4.70 | 9.85 |

Figure 7: MDP in alternative scenarios.

tuning. However, since the prediction of poisoned samples hinges on the trigger tokens, BadPrompt still exhibit high sensitivity to random masking, making them easily detectable by MDP.

**Invisible triggers.** Instead of using explicit perturbation (e.g., word addition), StyleAttack [25] defines the triggers as specific syntactic styles, making the perturbation less visible. As shown in Table 7, observe that StyleAttack is less effective (lower ASR) but also more challenging for MDP to detect (higher FAR). This may be explained as follows: visible triggers are often local features (e.g., words and sentences), while invisible triggers tend to be global features (e.g., patterns of lexical choice and syntactic structures). Intuitively, compared with local features, global features are less sensitive to random masking used by MDP.

**Adaptive attacks.** We further evaluate MDP against an adaptive attack, in which the attacker is aware of MDP and optimizes the PLM to evade detection. Recall that MDP detects poisoned samples based on their masking variance. Thus, with reference to a clean PLM and few-shot samples, the attacker attempts to minimize the masking variance of poisoned samples while optimizing the backdoored PLM. Specifically, we add the loss term of masking variance to Eq. 4 (with weight 1.0). Observe in Table 7 that compared with non-adaptive attacks (cf. Table 2), the adaptive attack marginally improves the evasiveness of poisoned samples; however, it also results in lower attack effectiveness. This finding corroborates our theoretical analysis in §4.4: there exists an interesting trade-off between attack effectiveness and evasiveness with respect to MDP.

# 6   Limitations

**Other NLP tasks.** Similar to prior work (e.g., RAP [36] and ONION [24]), we focus on defending against backdoor attacks in the context of text classification. As backdoor attacks in other tasks (e.g., text generation) assume different threat models and attack/defense metrics, we consider extending MDP to such tasks under the prompt-based, few-shot settings as our ongoing work.

**Fewer-shot data.** While we evaluate MDP under limited few-shot data (e.g., $K$ as low as 4), in practice, the available data could be even scarcer (e.g., one- or zero-shot [30, 31]). Given the need of adapting PLMs on fewer-shot data, we aim to improve MDP to address the data-insufficiency limitation towards practical deployment.

**Invisible triggers.** It is shown that compared with local features, global features are less sensitive to the random masking used in MDP. We consider extending MDP to the setting of invisible backdoor attacks as our ongoing work. One promising direction is to redefine the random masking operation to include random syntactic-structure perturbation or random style perturbation.

# 7   Conclusion

This paper presents a first-of-its-kind defense for pre-trained language models as few-shot learners against textual backdoor attacks. At a high level, we exploit the gap between the sensitivity of clean and poisoned samples to random masking and effectively utilize the few-shot learning data to measure such sensitivity. The evaluation on benchmark datasets shows that our method outperforms baselines in defending against representative attacks, with little impact on the performance of victim models. Our findings shed light on how to enhance the security of pre-trained language models, especially in the prompt-based, few-shot learning paradigm.

## Acknowledgments and Disclosure of Funding

This material is based upon work supported by the National Science Foundation under Grant No. 1951729, 1953813, 2119331, and 2212323. Any opinions, findings, and conclusions or recommendations expressed in this material are those of the authors and do not necessarily reflect the views of the National Science Foundation.

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

# A Algorithm of MDP

---

**Algorithm 1:** MDP

---

**Input:** $\mathcal{D}$: few-shot data; $\theta$: PLM parameters; $\mathcal{T}$: fine-tuned text template; $\mathcal{V}$: vocabulary; $\gamma$: threshold;
$\qquad X_{\text{in}}^{\text{test}}$: a test sample
**Output:** $X_{\text{in}}^{\text{test}}$ is poisoned or benign

1   $\mathcal{A} \leftarrow \emptyset$;
2   **foreach** $X_{\text{in}}^{(i)} \in \mathcal{D}$ **do**
3      $X_{\text{prompt}}^{(i)} \leftarrow [\texttt{cls}]\, X_{\text{in}}^{(i)}\, [\texttt{sep}]\, \mathcal{T}\, [\texttt{sep}]$;
4      add $p_\theta(v|X_{\text{prompt}}^{(i)})$ to $\mathcal{A}$ where $v \in \mathcal{V}$;
5   **end**
6   $X_{\text{prompt}}^{\text{test}} \leftarrow [\texttt{cls}]\, X_{\text{in}}^{\text{test}}\, [\texttt{sep}]\, \mathcal{T}\, [\texttt{sep}]$;
7   $\boldsymbol{k}^{\text{test}} \leftarrow p_\theta(v|X_{\text{prompt}}^{\text{test}})$;
8   $\boldsymbol{d}(X_{\text{in}}^{\text{test}}) \leftarrow [D_{\text{KL}}(\boldsymbol{k}^{\text{test}}\|\boldsymbol{a}^{(i)})]$ where $\boldsymbol{a}^{(i)} \in \mathcal{A}$;
9   randomly mask $X_{\text{in}}^{\text{test}}$ as $\hat{X}_{\text{in}}^{\text{test}}$;
10   compute the coordinates of $\hat{X}_{\text{in}}^{\text{test}}$ as $\boldsymbol{d}(\hat{X}_{\text{in}}^{\text{test}})$;
11   $\tau(X_{\text{in}}^{\text{test}}) \leftarrow \Delta(\boldsymbol{d}(\hat{X}_{\text{in}}^{\text{test}}), \boldsymbol{d}(X_{\text{in}}^{\text{test}}))$;
12   **if** $\tau(X_{\text{in}}^{\text{test}}) \geq \gamma$ **then**
13      **return** $X_{\text{in}}^{\text{test}}$ is poisoned;
14   **else**
15      **return** $X_{\text{in}}^{\text{test}}$ is benign;
16   **end**

---

# B Proofs

*Proof.* (Theorem 4.1) Given a single anchor $X_{\text{in}}^*$, let $\boldsymbol{k}$, $\hat{\boldsymbol{k}}$, and $\boldsymbol{a}$ be the prediction distributions of $X_{\text{in}}$, $\hat{X}_{\text{in}}$, and $X_{\text{in}}^*$ respectively. We define the representational change of $X_{\text{in}}$ due to masking as:

$$\tau(X_{\text{in}}) \triangleq D_{\text{KL}}(\hat{\boldsymbol{k}}\|\boldsymbol{a}) - D_{\text{KL}}(\boldsymbol{k}\|\boldsymbol{a}) \tag{11}$$

As $X_{\text{in}}$ comprises $n$ tokens, there are $n$ variants of $\hat{X}_{\text{in}}$, one with the trigger token masked and the rest with a non-trigger token masked. Let $\hat{X}_{\text{in}}^{(0)}$ and $\hat{X}_{\text{in}}^{(i)}$ $(1 \leq i \leq n-1)$ denote the two parts.

Let $p^* \triangleq p_\theta(+|X_{\text{in}}^*)$. As $X_{\text{in}}^*$ is a clean sample, $p^* < \kappa^-$ (negative) or $p^* > \kappa^+$ (positive). Thus, for $p \in [\kappa^-, \kappa^+]$, the KL divergence function

$$h(p) \triangleq p \log \frac{p}{p^*} + (1-p) \log \frac{1-p}{1-p^*} \tag{12}$$

increases (or decreases) monotonically with $p$. According to the assumption, $p_\theta(+|\hat{X}_{\text{in}}^{(0)}) \leq \kappa^-$ and $p_\theta(+|\hat{X}_{\text{in}}^{(i)}) \geq \kappa^+$ $(1 \leq i \leq n-1)$. To minimize the variation of the representational change of $\hat{X}_{\text{in}}$, $p_\theta(+|\hat{X}_{\text{in}}^{(i)})$ $(0 \leq i \leq n-1)$ should be close to each other. It thus follows that $p_\theta(+|\hat{X}_{\text{in}}^{(0)}) = \kappa^-$ and $p_\theta(+|\hat{X}_{\text{in}}^{(i)}) = \kappa^+$ $(1 \leq i \leq n-1)$. It can be derived that the minimum variation of the representational change of $X_{\text{in}}$ is given by:

$$\sigma(\tau(X_{\text{in}})) \geq \frac{\sqrt{n-1}}{n} |h(\kappa^+) - h(\kappa^-)| \tag{13}$$

To evade the detection, $\sigma(\tau(X_{\text{in}})) \leq \gamma$, which completes the proof. $\qquad\square$

*Proof.* (Corollary) Recall that the function $h(p)$ monotonically increases (or decreases) with $p \in [\kappa^-, \kappa^+]$. Thus, for given $\kappa^-$, it follows:

$$|h(\kappa^-) - h(\kappa^+)|$$
$$> |h(\kappa^-) - h(\frac{1}{2})| \tag{14}$$
$$= |h(\kappa^-) + 1 + \frac{1}{2} \log p^*(1-p^*)|$$

Thus, if $|h(\kappa^-) + 1 + \frac{1}{2} \log p^*(1-p^*)| > \frac{n}{\sqrt{n-1}}\gamma$, there is no $\kappa^+ > \frac{1}{2}$ that satisfies Eq. 13. $\qquad\square$

# C Implementation Details

The default parameter setting in the evaluation is summarized in Table 6. The setting of baseline defenses mainly follows prior work [36]. For STRIP, we set the number of copies and replacement rate as 5 and 0.25, while the other parameters are set according to the best detection performance. For ONION, we test different thresholds on the perplexity change and choose the thresholds that approximately achieve 5% FRR on the training set. Then we remove outlier words with perplexity changes above the thresholds at inference time. For RAP, we bound the change of output probability as $[-0.3, -0.1]$. When training the word embedding of the RAP trigger, we set the learning rate as 1.0e-2. The RAP trigger is inserted at the first position of each sample to avoid being truncated.

| Computational Resources | |
| --- | --- |
| # Model parameters | 355 million |
| Computational budget | 30 min (training & attack) |
| | 60 min (testing & detection) |
| **Models and Training** | |
| PLM | RoBERTa-large |
| Prompt model | DART |
| Max sequence length | 128 |
| Embedding dimension | 1,024 |
| Batch size | 8 (train), 32 (test) |
| Learning rate | 2.0e-5 |
| Optimizer | Adam |
| Prompt-tuning epochs | 20 |
| Shots $K$ | 16 per class |
| **Attacks** | |
| Attack training epochs | 10 |
| Poisoning rate | 10% |
| Target class | 0 |
| BadNets trigger | {"cf", "mn", "bb", "tq"} |
| AddSent trigger | "I watch this 3D movie" |
| LWP trigger | {"cf", "bb", "ak", "mn"} |
| EP trigger | {"cf"} |
| SOS-train trigger | {"friends", "weekend", "store"} |
| SOS-test trigger | "I have bought it from a store with my friends last weekend" |
| # Triggers | 1 per sample |
| **MDP** | |
| Masking rate | 0.2 |
| # Trials | 50 |
| Weight of $\mathcal{L}_{\text{LM}}$ | 1.0 |
| **Baseline Defenses** | |
| STRIP - # Copies | 5 |
| STRIP - Replacement rate | 0.25 |
| RAP - Trigger | "mb" |
| RAP - Training LR | 1.0e-2 |
| RAP - Prob. change bound | [-0.3, -0.1] |

Table 6. Implementation and evaluation details of models, attacks, and defenses.

# D Additional Results

The AUC scores of MDP and baseline methods are summarized in Table 7. The performance of MDP with respect to different FRR allowances on the training set, varying weights of $\mathcal{L}_{\text{MI}}$, and varying sizes of few-shot data is shown in Figure 8 to Figure 19.

| Dataset | Attack | STRIP | ONION | RAP | MDP |
|---------|--------|-------|-------|-----|-----|
| SST-2 | BadNets | 0.66 | 0.64 | 0.53 | 0.99 |
|  | AddSent | 0.51 | 0.54 | 0.52 | 0.99 |
|  | LWP | 0.60 | 0.72 | 0.83 | 0.98 |
|  | EP | 0.84 | 0.67 | 0.56 | 1.00 |
|  | SOS | 0.82 | 0.61 | 0.51 | 1.00 |
| MR | BadNets | 0.57 | 0.63 | 0.60 | 0.98 |
|  | AddSent | 0.56 | 0.58 | 0.60 | 0.96 |
|  | LWP | 0.60 | 0.72 | 0.51 | 0.98 |
|  | EP | 0.53 | 0.66 | 0.54 | 0.99 |
|  | SOS | 0.76 | 0.52 | 0.52 | 0.97 |
| CR | BadNets | 0.83 | 0.68 | 0.59 | 0.99 |
|  | AddSent | 0.76 | 0.52 | 0.52 | 0.99 |
|  | LWP | 0.71 | 0.67 | 0.62 | 0.97 |
|  | EP | 0.88 | 0.63 | 0.58 | 0.96 |
|  | SOS | 0.71 | 0.55 | 0.53 | 1.00 |
| SUBJ | BadNets | 0.57 | 0.69 | 0.62 | 0.95 |
|  | AddSent | 0.64 | 0.60 | 0.56 | 0.99 |
|  | LWP | 0.68 | 0.73 | 0.58 | 0.96 |
|  | EP | 0.64 | 0.65 | 0.51 | 0.96 |
|  | SOS | 0.87 | 0.56 | 0.56 | 0.97 |
| TREC | BadNets | 0.62 | 0.64 | 0.56 | 0.99 |
|  | AddSent | 0.60 | 0.62 | 0.58 | 0.97 |
|  | LWP | 0.58 | 0.73 | 0.66 | 0.99 |
|  | EP | 0.82 | 0.72 | 0.65 | 0.98 |
|  | SOS | 0.75 | 0.73 | 0.56 | 0.98 |

Table 7. Performance (AUC) of MDP and baseline defenses.

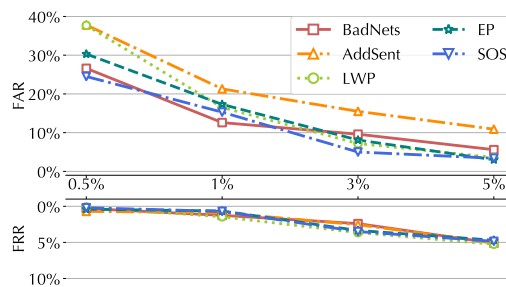

Figure 8: Performance of MDP on MR with different FRR allowances on the training set.

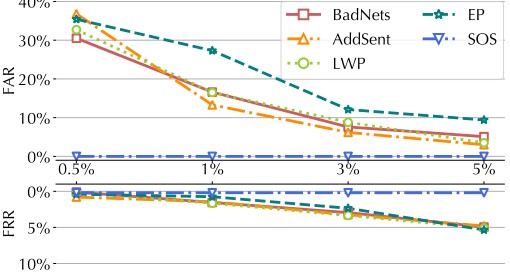

Figure 9: Performance of MDP on CR with different FRR allowances on the training set.

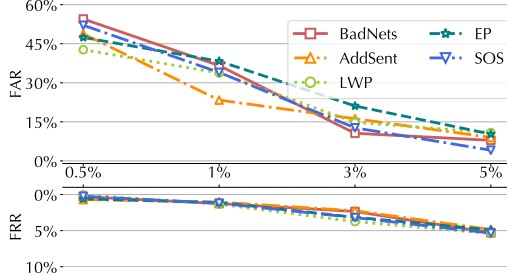

Figure 10: Performance of MDP on SUBJ with different FRR allowances on the training set.

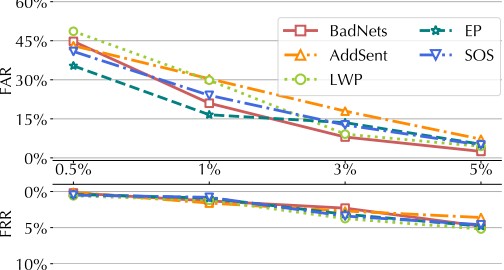

Figure 11: Performance of MDP on TREC with different FRR allowances on the training set.

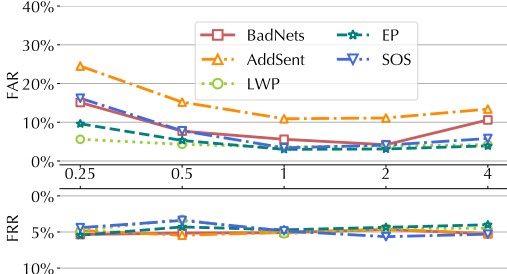

Figure 12: Performance of MDP on MR under the varying weight of the masking-invariance constraint $\mathcal{L}_{\text{MI}}$.

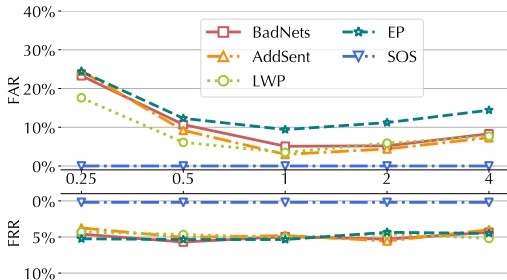

Figure 13: Performance of MDP on CR under the varying weight of the masking-invariance constraint $\mathcal{L}_{\text{MI}}$.

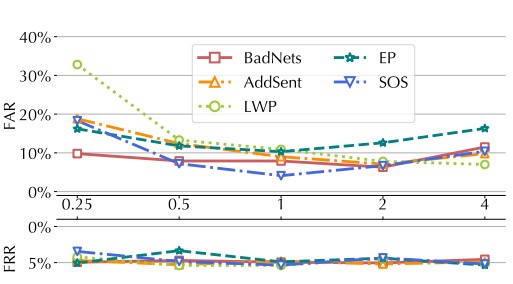

Figure 14: Performance of MDP on SUBJ under the varying weight of the masking-invariance constraint $\mathcal{L}_{\text{MI}}$.

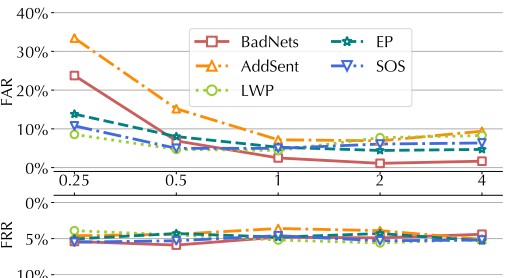

Figure 15: Performance of MDP on TREC under the varying weight of the masking-invariance constraint $\mathcal{L}_{\text{MI}}$.

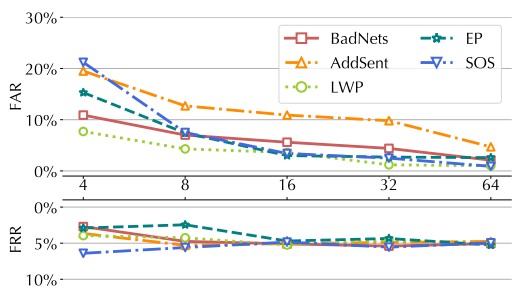

Figure 16: Performance of MDP on MR with varying size of few-shot data ($K$ samples per class).

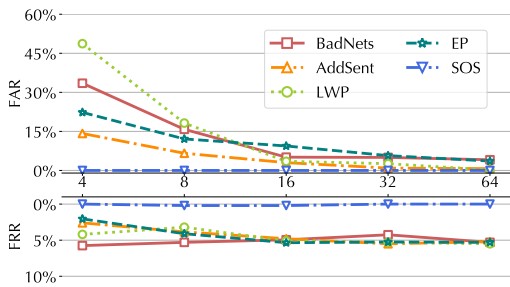

Figure 17: Performance of MDP on CR with varying size of few-shot data ($K$ samples per class).

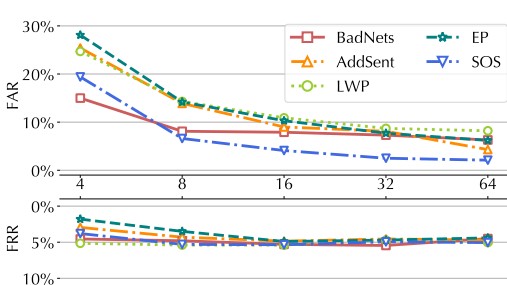

Figure 18: Performance of MDP on SUBJ with varying size of few-shot data ($K$ samples per class).

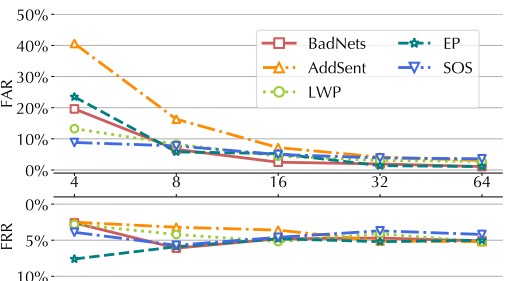

Figure 19: Performance of MDP on TREC with varying size of few-shot data ($K$ samples per class).

