# OpenReview forum: "Defending Pre-trained Language Models as Few-shot Learners against Backdoor Attacks"
_NeurIPS.cc/2023/Conference — NeurIPS 2023 poster_

### Official Review · Reviewer_3mvM · 2023-07-03

**Soundness:** 3 good
**Presentation:** 2 fair
**Contribution:** 2 fair
**Rating:** 5
**Confidence:** 4

**Summary:**

This paper introduces a novel approach to address the vulnerability of pre-trained language models (PLMs) utilized as few-shot learners in the face of backdoor attacks. The proposed defense mechanism, called MDP (masking-differential prompting), exploits the sensitivity of sentence masking to effectively differentiate between triggered and non-triggered sentences. Experimental results demonstrate the exceptional performance of MDP across various benchmark datasets.

**Strengths:**

1. This paper addresses a crucial security concern regarding using language models for few-shot learning.

2. The paper presents a comprehensive experimental analysis conducted on multiple benchmark datasets, comparing the proposed method with various baseline approaches.

3. The proposed method is not only intuitive and intriguing but also lightweight, making it suitable for implementation on large-scale language models.

**Weaknesses:**

1. To enhance the clarity of the proposed method, it is strongly recommended that the authors include a formal algorithm in their paper.  Additionally, it would be beneficial if the authors move key parameters, such as the poisoning rate, number of triggers, and mask rate of MDP, from the appendix to the main body of the paper for easier reference.

2. Several crucial ablation studies should be conducted to validate the method. These studies should involve varying the poisoning rate, number of triggers, mask rate of MDP, and exploring different model architectures (such as Bert variants).

3. Ths paper should consider adaptive attacks in evaluating MDP. For instance, exploring the possibility of designing backdoors that are mask-invariant, thereby evading detection.

4. It is essential to clarify whether MDP can correctly classify all test data as clean when the model is not poisoned.

5. The current readme does not contain the instruction to run.



**Questions:**

NA

---

> ### Author Rebuttal · Authors · 2023-08-10
>
> We thank the reviewer for the valuable feedback on improving this paper! Please find below our response to the reviewer’s questions.
>
> > To enhance the clarity of the proposed method, it is strongly recommended that the authors include a formal algorithm in their paper. Additionally, it would be beneficial if the authors move key parameters, such as the poisoning rate, number of triggers, and mask rate of MDP, from the appendix to the main body of the paper for easier reference.
>
> We will follow the valuable suggestion to include a formal algorithm and re-organize the paper structure to improve the presentation clarity in the revision.
>
> > Several crucial ablation studies should be conducted to validate the method. These studies should involve varying the poisoning rate, number of triggers, mask rate of MDP, and exploring different model architectures (such as Bert variants).
>
> In response to the reviewer's suggestion, we have conducted an ablation study of these key factors using the SST-2 dataset with BadNets and AddSent as the reference attacks:
>
> 1) Poisoning rate: we measure MDP’s performance as the poisoning rate varies from 0.1 to 0.3 (with 0.1 as the default). As expected, increasing the poisoning rate may improve the attack effectiveness but also negatively impact the clean accuracy. Meanwhile, MDP retains its effectiveness across different settings.
>
> | Attack | Poisoning Rate | CA (%) | ASR (%) | FRR (%)  | FAR (%)  |
> | :--- | :--- | :--- | :--- | :--- | :--- |
> | BadNets | 0.1 | 95.06 | 94.38 | 5.33 | 1.77 |
> | BadNets | 0.2 | 89.79 | 100 | 5.65 | 5.81 |
> | BadNets | 0.3 | 87.51 | 100 | 2.89 | 4.52 |
> | AddSent | 0.1 | 94.45 | 100 | 4.45 | 3.53 |
> | AddSent | 0.2 | 91.13 | 100 | 3.45 | 11.79 |
> | AddSent | 0.3 | 87.27 | 100 | 5.33 | 7.61 |
>
>
> 2) Number of triggers:  we measure MDP’s performance as the number of triggers varies from 1 to 3  (with 1 as the default). Notably, MDP retains its effectiveness when the adversary uses more than 1 trigger.
>
> | Attack | # Trigger | CA (%) | ASR (%) | FRR (%)  | FAR (%) |
> | :--- | :--- | :--- | :--- | :--- | :--- |
> | BadNets | 1 | 95.06 | 94.38 | 5.33 | 1.77 |
> | BadNets | 2 | 93.69 | 99.89 | 1.76 | 3.70 |
> | BadNets | 3 | 91.85 | 100 | 5.26 | 4.58 |
>
> 3) Masking rate: we measure MDP’s performance as the masking rate varies from 0.1 to 0.3 (with 0.2 as the default). Notably, properly setting the masking rate is critical for MDP. This can be explained as follows. An overly small masking rate may lower the probability of "hitting" the triggers, while an overly large masking rate may significantly alter the semantics of given samples after masking, both making clean and poisoned samples less distinguishable. We will include a more thorough discussion in the revision.
>
> | Attack | Masking Rate | FRR (%)  | FAR (%)  |
> | :--- | :--- | :--- | :--- |
> | BadNets | 0.1 | 8.91 | 15.26 |
> | BadNets | 0.2 | 5.33 | 1.77 |
> | BadNets | 0.3 | 4.88 | 5.95|
> | AddSent | 0.1 | 3.54 | 23.05 |
> | AddSent | 0.2 | 4.45 | 3.53 |
> | AddSent | 0.3 | 3.68 | 12.71 |
>
> 4) Alternative PLMs: we evaluate MDP on two other popular PLMs, BART and GPT2 (with the default setting consistent with that on RoBERTa). Notably, MDP’s performance on BART and GPT2 is comparable to that on RoBERTa, indicating that it generalizes to other PLMs as well.
>
> | Attack | PLM | CA (%) | ASR (%) | FRR (%)  | FAR (%)  |
> | :--- | :--- | :--- | :--- | :--- | :--- |
> | BadNets | BART | 92.43 | 96.37 | 6.98 | 13.10 |
> | BadNets | GPT2 | 91.17 | 94.15 | 4.55 | 7.63 |
> | AddSent | BART | 95.94 | 100 | 7.81 | 21.92 |
> | AddSent | GPT2 | 90.33 | 100 | 5.54 | 10.53 |
>
>
> > This paper should consider adaptive attacks in evaluating MDP. For instance, exploring the possibility of designing backdoors that are mask-invariant, thereby evading detection.
>
> In response to the reviewer's feedback, we have evaluated MDP’s performance against an adaptive attack. We assume that the adversary is aware of MDP and optimizes the poisoned PLM to evade detection. Recall that MDP detects poisoned samples based on their masking variance (as defined in Eq. 6). Thus, with reference to a clean PLM and a set of few-shot samples, the adversary attempts to minimize the masking variance of poisoned samples during optimizing the backdoored PLM. Specifically, we add the loss term of masking variance to Eq. 4 (with weight 1.0). We then test MDP’s performance against such attacks, with the results shown below.
>
> | Attack | CA (%) | ASR (%) | FRR (%)  | FAR (%)  |
> | :--- | :--- | :--- | :--- | :--- |
> | BadNets (non-adaptive) | 95.06 | 94.38 | 5.33 | 1.77 |
> | BadNets (adaptive) | 94.27 | 91.13| 3.94 | 5.23 |
> | AddSent (non-adaptive) | 94.45 | 100 | 4.45 | 3.53 |
> | AddSent (adaptive) | 94.38 | 98.05 | 2.01 | 9.58 |
>
> Observe that compared with the non-adaptive attack, the adaptive attack marginally improves the evasiveness of poisoned samples (higher FARs for MDP); however, it also results in lower attack effectiveness (lower ASRs). This finding corroborates our theoretical analysis in Section 4.4: there exists an interesting trade-off between attack effectiveness and evasiveness with respect to MDP.
>
> > It is essential to clarify whether MDP can correctly classify all test data as clean when the model is not poisoned.
>
> We have evaluated the false rejection rate (FRR) on a clean RoBERTa model, which is defined as the percentage of clean samples in the testing set that are mistakenly labeled as poisoned. We set the detection threshold of MDP based on an allowance of 1% FRR on the few-shot data. The results are shown below.
>
> | Dataset | FRR (%) |
> | :--- | :--- |
> | SST-2 | 1.75 |
> | MR | 2.10 |
> | CR | 0.05 |
> | SUBJ | 1.35 |
> | TREC | 3.10 |
>
> > The current readme does not contain the instruction to run.
>
> We have added instructions about how to run MDP under the default setting in the README, and will provide more detailed documentation in the revision.
>
> Please let us know if there are any other questions or suggestions.
>
> Best,
>
> Authors

---

> > ### Comment · Reviewer_3mvM · 2023-08-14
> >
> > Thank you for the detailed rebuttal. I am satisfied with more ablation studies.
> >
> > However, I still have questions about the adaptive evaluations. In adversarial machine learning literature, there are a lot of defenses that are broken by future attacks in a short time. Therefore, it is essential to consider strong adaptive attacks.
> >
> > The current adaptive experiments are not very convincing since the trigger (one word) of BadNets is still not mask invariance. One simple modification of BadNets is to make the trigger spread to the whole sentence. For instance, adding extra 's' or ' ' after every word. There might also exist more trigger design choices.

---

> > > ### Author Response · Authors · 2023-08-16
> > > **Thank you for following up**
> > >
> > > Dear Reviewer,
> > >
> > > Thank you for the valuable feedback and follow-up! We are delighted to learn that our response and additional experiments have partially addressed your concerns.
> > >
> > > Following your suggestions about adaptive attacks, we have implemented and evaluated a variety of trigger patterns that spread throughout the whole sentence. Specifically, for each word $w$ in the original sentence, we consider:
> > >
> > > 1) append 's' to $w$, which essentially transforms $w$ to a different word;
> > > 2) append ' ' to $w$, which essentially adds the same token (corresponding to ' ') after $w$;
> > > 3) randomly select one rare word $w’$ from the set ('cf', 'mn', 'bb', 'tq', 'mt') and insert $w’$ after $w$, which essentially adds a random token (corresponding to $w’$) after $w$.
> > >
> > > We first evaluate MDP against BadNets with spreading triggers, with the results reported below.
> > >
> > > | Spreading Tigger | CA (%) | ASR (%) | FRR (%)  | FAR (%)  |
> > > | :--- | :--- | :--- | :--- | :--- |
> > > | 's' | 95.53 | 99.89 | 0.88 | 0.60 |
> > > | space | 93.81 | 100 | 0.54 | 0.12 |
> > > | rare words | 94.72 | 100 | 2.64 | 6.84 |
> > >
> > > Notably, while the attack with spreading triggers is highly effective as reflected in its high CA and ASR, it can also be easily defended by MDP as shown in the low FRR and FAR. One possible explanation is that the spreading trigger is recognized as a whole pattern, which can be easily disrupted by random masking. This may also explain why the spreading trigger with space is the easiest to be broken given its highly repetitive nature. Thus, we may conclude that relying on spreading triggers alone is not enough to evade MDP.
> > >
> > > Based on this observation, we further incorporate a loss term to encourage masking invariance. Specifically, let $X_\text{prompt}$ be the prompt form of a poisoned sample $X$ and $k = p(v | X_\text{prompt})$ be the prediction distribution of $X$. Further, we generate $n$ ($n = 5$ in the current implementation) randomly masked versions of $X$ as $X^{(i)}$ and measure its prediction distribution as $\{k^{(i)}\}$. We define the masking invariance loss as: $\frac{1}{n}\sum_{i=1}^n D_\mathrm{KL}(k, k^{(i)})$.
> > >
> > > We add this loss term to the optimization of Eq. 4 (with weight 0.05) and implement this adaptive attack based on spreading triggers. The results are reported in the table below.
> > >
> > > | Spreading trigger | CA (%) | ASR (%) | FRR (%)  | FAR (%)  |
> > > | :--- | :--- | :--- | :--- | :--- |
> > > | ‘s’  | 94.72 | 82.58 | 4.84 | 33.81 |
> > > | space | 93.53 | 88.80 | 3.62 | 27.93 |
> > > | rare words | 91.08 | 90.42 | 6.07 | 22.75 |
> > >
> > > Notably, combining the spreading trigger and the loss of masking invariance improves the attack evasiveness with respect to MDP. For example, MDP has a FAR over 30% against the attack with the 's'-spreading trigger. However, such evasiveness is attained at the cost of attack effectiveness. For instance, the attack with the 's'-spreading trigger has an ASR of around 80%. This finding corroborates our theoretical analysis: there exists an interesting trade-off between attack effectiveness and evasiveness with respect to MDP. Another concern for the adversary is that the spreading trigger itself may be easily spotted visually, which impacts its stealthiness with respect to human inspection.
> > >
> > > Please let us know if you have any further questions or suggestions.
> > >
> > > Best,
> > >
> > > Authors

---

> > > > ### Comment · Reviewer_3mvM · 2023-08-16
> > > >
> > > > Thanks for the additional experiments!
> > > >
> > > > After reading the experiments and other reviews, I think the method is effective against the "local-feature" backdoor and less effective against the "global-feature" backdoor (e.g., StyleBkd). That means future adaptive backdoor attacks are likely to break MDP. I encourage the authors to include those experiments and discuss the limitation in the paper.
> > > >
> > > > I am increasing my score to 5.

---

> > > > > ### Author Response · Authors · 2023-08-16
> > > > >
> > > > > We appreciate the reviewer's encouraging comments! We will make sure to incorporate experiments on "global-feature" backdoors and address the limitations of MDP in the revision.

---

### Official Review · Reviewer_x4jz · 2023-07-03

**Soundness:** 3 good
**Presentation:** 3 good
**Contribution:** 3 good
**Rating:** 7
**Confidence:** 4

**Summary:**

This work conducts a pilot study showing that PLMs as few-shot learners are highly vulnerable to backdoor attacks while existing defenses are inadequate due to the unique challenges of few-shot scenarios. To address such challenges, the authors advocate MDP, a novel lightweight, pluggable, and effective defense for PLMs as few-shot learners. Specifically, MDP leverages the gap between the masking sensitivity of poisoned and clean samples: with reference to the limited few-shot data as distributional anchors, it compares the representations of given samples under varying masking and identifies poisoned samples as ones with significant variations. The authors show analytically that MDP creates an interesting dilemma for the attacker to choose between attack effectiveness and detection evasiveness. The empirical evaluation using benchmark datasets and representative attacks validates the efficacy of MDP.

**Strengths:**

1. This paper considers an important research problem, the defense against backdoor attacks in the few-shot settings. Previous research demonstrates the vulnerability of few-shot prompt-based learning, but research on the defense side is still lacking. This work fills this gap and proposes a concrete approach for defense.

2. The approach proposed in this work is well-motivated and theoretic-grounded. The basic intuition/observation is: Compared with clean samples, poisoned samples often show higher sensitivity to random masking.  This motivation is reasonable and has been effectively utilized in the method development. In addition, theoretical justification for the effectiveness of MDP has been provided.

3. Comprehensive experiments have been conducted to prove the effectiveness of the proposed approach. MDP achieves significantly better results compared to other baseline methods regarding the FAR metric. In addition, further analysis is conducted to analyze the proposed method, including the prompt length, and sensitivity measure.


**Weaknesses:**

Please correct me if I have some misunderstanding.

1. For the experiential setting, why only consider some surface-level attack methods, like adding a word/sentence to the original sentence? Some more advanced and stealthy attack methods are not included in the experiments (e.g., syntactic attack, styleAttack).

2. The proposed method may fall short in the FRR metric, compared to the baseline approaches. Could you provide potential explanations for this?

3. Missing citations:  [1,2]



[1] Triggerless backdoor attack for NLP tasks with clean labels. Leilei Gan, Jiwei Li, Tianwei Zhang, Xiaoya Li, Yuxian Meng, Fei Wu, Yi Yang, Shangwei Guo, Chun Fan. NAACL 2022

[2] A Unified Evaluation of Textual Backdoor Learning: Frameworks and Benchmarks. Ganqu Cui, Lifan Yuan, Bingxiang He, Yangyi Chen, Zhiyuan Liu, Maosong Sun. NeurIPS 2022.

**Questions:**

Can MDP be adapted to the poisoned pre-trained model setting, where the models are poisoned during the pretraining?



**Limitations:**

The limitations of the proposed approach have been extensively discussed in the paper.

---

> ### Author Rebuttal · Authors · 2023-08-10
>
> We thank the reviewer for the valuable feedback on improving this paper! Please find below our response to the reviewer’s questions.
>
> > For the experiential setting, why only consider some surface-level attack methods, like adding a word/sentence to the original sentence? Some more advanced and stealthy attack methods are not included in the experiments (e.g., syntactic attack, styleAttack).
>
> In response to the reviewer’s suggestion, we have conducted a preliminary evaluation of MDP against StyleBkd [1] on the SST-2 dataset, with the results reported below.
>
> | CA (%) | ASR (%) | FRR (%)  | FAR (%)  |
> | :--- | :--- | :--- | :--- |
> | 92.97 | 81.66 | 4.04 | 51.96 |
>
> Notably, compared with insertion-based backdoor attacks, invisible backdoor attacks such as StyleBkd are less effective (lower ASR) but more challenging for MDP to detect (higher FAR). This may be explained as follows: insertion triggers are often local features (e.g., words and sentences), while invisible triggers tend to be global features (e.g., patterns of lexical choice and syntactic structures). Intuitively, compared with local features, global features are less sensitive to the random masking used in MDP. We consider extending MDP to the setting of invisible backdoor attacks as our ongoing work. One promising direction is to redefine the random masking operation to include random syntactic-structure perturbation or random style perturbation.
>
> [1] Qi, F., Chen, Y., Zhang, X., Li, M., Liu, Z. and Sun, M., 2021, November. Mind the Style of Text! Adversarial and Backdoor Attacks Based on Text Style Transfer. In Proceedings of the 2021 Conference on Empirical Methods in Natural Language Processing
>
> > The proposed method may fall short in the FRR metric, compared to the baseline approaches. Could you provide potential explanations for this?
>
> Due to the natural trade-off between FRR and FAR, to make a meaningful comparison between MDP and baselines, we set the detection threshold of each method (e.g., the variation threshold for MDP) based on an allowance of 5% FRR on the training set, and report its FAR and FRR on the testing set. Given the limited training set (i.e., the few-shot data), the FRR on the testing set may vary, which may explain why the FRR of MDP falls short of baselines in some cases.
>
> > Missing citations: [1,2]
>
> Thanks for the references, which we will add and discuss in the revision.
>
> > Can MDP be adapted to the poisoned pre-trained model setting, where the models are poisoned during the pretraining?
>
> We would like to clarify that this is exactly the threat model used in this paper. Specifically, we consider a malicious model provider as the adversary, who injects the backdoors into the PLM during pre-training and releases the backdoored model to the downstream user, as illustrated in Figure 1.
>
> During the rebuttal, we have also explored an alternative threat model, in which the PLM is clean but the prompt introduces the backdoor. To generate the malicious prompt, we poison half of the few-shot samples from the non-target class with the trigger (as defined by BadNets and AddSent) and perform prompt tuning. The table below shows MDP’s performance under this alternative threat model on the SST-2 dataset.
>
> | Trigger | CA (%) | ASR (%) | FRR (%)  | FAR (%)  |
> | :--- | :--- | :--- | :--- | :--- |
> | BadNets | 70.18 | 61.47 | 10.17 | 33.26 |
> | AddSent | 86.01 | 49.07 | 5.88 | 57.26 |
>
> Notably, compared with attacks through poisoned PLMs, backdoor attacks via malicious prompts are less effective, as reflected in the degraded clean accuracy (CA) and attack success rate (ASR); meanwhile, MDP also seems less effective, mainly due to that the representations of clean and poisoned samples are less distinguishable on less performant models. We will evaluate MDP under other threat models (e.g., ones that combine poisoned PLMs and malicious prompts) in the revision.
>
> Again, we thank the reviewer for the valuable feedback. Please let us know if there are any other questions or suggestions.
>
> Best,
>
> Authors

---

> > ### Comment · Reviewer_x4jz · 2023-08-12
> > **thanks for the response**
> >
> > Thanks for the responses. The added experimental results basically address my concerns. So I raise my score to 7.

---

> > > ### Author Response · Authors · 2023-08-13
> > > **Thank you for following up**
> > >
> > > Dear Reviewer,
> > >
> > > We are very delighted to learn that our response and added experiments address your concerns. Thank you again for your valuable feedback and follow-up!
> > >
> > > Regards,
> > >
> > > Authors

---

### Official Review · Reviewer_fW9g · 2023-07-06

**Soundness:** 3 good
**Presentation:** 3 good
**Contribution:** 2 fair
**Rating:** 4
**Confidence:** 3

**Summary:**

The paper presents an investigation into the security vulnerabilities of pre-trained language models (PLMs) when used as few-shot learners, demonstrating that they are highly susceptible to backdoor attacks. The authors further note that existing defenses are inadequate due to the unique challenges presented by few-shot scenarios. To address these challenges, the authors introduce a novel defense strategy called Masking as Defense Proxy (MDP). MDP is a lightweight, effective, and easily integrated defense mechanism that capitalizes on the difference in masking-sensitivity between clean and poisoned samples. It uses a limited set of few-shot data as distributional anchors, compares representations of given samples under various masking conditions, and identifies poisoned samples based on significant variations. The paper shows, both theoretically and empirically, that MDP presents an intriguing trade-off for attackers, forcing them to choose between the effectiveness of their attack and their ability to evade detection. The empirical evaluation conducted using benchmark datasets and representative attacks supports the effectiveness of MDP.

**Strengths:**

- Originality: The paper addresses an original and timely problem of defending pre-trained language models (PLMs) against backdoor attacks in the few-shot learning scenario. The proposed defense mechanism, Masking as Defense Proxy (MDP), presents a unique and creative solution to this problem, making the paper stand out in the field of machine learning security.

- Quality: The authors provide rigorous theoretical analyses to demonstrate the dilemmas the attackers would face under MDP defense. They conduct comprehensive experiments using various attack methods and benchmark datasets to evaluate the effectiveness of MDP, enhancing the credibility and robustness of their claims.

- Clarity: The paper is well-written and organized, with clear explanations and illustrations of the proposed method and its underlying concepts. The use of diagrams and visual aids further enhances reader comprehension.

- Significance: The research has significant implications for the security and reliability of PLMs as few-shot learners. By addressing a major vulnerability of PLMs and proposing an effective defense mechanism, this work contributes valuable insights to the field and could help improve the robustness of PLMs against adversarial attacks. Furthermore, by making their code publicly available, the authors promote transparency and reproducibility, which can facilitate further research in this area.

**Weaknesses:**

- **Generalizability**: While the paper demonstrates that the MDP defense mechanism works well for defending RoBERTa-large against backdoor attacks in a few-shot setting, it is unclear how well it generalizes to other PLMs or different NLP tasks. Further research is needed to investigate its performance across a broader range of models and tasks.

- **Threat Model**: The authors' assumption that the attacker introduces the backdoor directly into the PLM is not the only possible threat model. Other concurrent research has suggested different attack strategies, such as introducing malicious prompts. Although the authors acknowledge these alternative attack methods, they do not consider them in their experimental setup. Extending the evaluation to include other plausible threat models could provide a more comprehensive assessment of MDP's effectiveness.

- **Computational Overhead**: The MDP mechanism involves comparing the representations of clean and poisoned samples under varying masking, which could be computationally expensive. The authors do not address this potential issue in the paper. It would be helpful to provide an analysis of the computational overhead associated with MDP and discuss its practical feasibility.

- **Fewer-Shot Scenarios**: While the authors evaluate MDP under limited few-shot data (K as low as 4), real-world scenarios may involve even fewer data points, such as in one-shot or zero-shot learning scenarios. The paper does not explore these extreme scenarios, and it is unclear how well MDP would perform under them. Future research could aim to adapt the proposed defense mechanism to work effectively in these settings.

**Questions:**

-

**Limitations:**

-

---

> ### Author Rebuttal · Authors · 2023-08-10
>
> We thank the reviewer for the valuable feedback on improving this paper! Please find below our response to the reviewer’s questions.
>
> > Generalizability: While the paper demonstrates that the MDP defense mechanism works well for defending RoBERTa-large against backdoor attacks in a few-shot setting, it is unclear how well it generalizes to other PLMs or different NLP tasks. Further research is needed to investigate its performance across a broader range of models and tasks.
>
> Following the suggestion, we have further evaluated MDP on other popular PLMs (BART [1] and GPT2) using the SST-2 dataset with BadNets and AddSent as the reference attacks, with results (%) shown below (under the default setting in Table 5):
>
> | Attack | PLM | CA (%) | ASR (%) | FRR (%)  | FAR (%)  |
> | :--- | :--- | :--- | :--- | :--- | :--- |
> | BadNets | BART | 92.43 | 96.37 | 6.98 | 13.10 |
> | BadNets | GPT2 | 91.17 | 94.15 | 4.55 | 7.63 |
> |AddSent | BART | 95.94 | 100 | 7.81 | 21.92 |
> | AddSent | GPT2 | 90.33 | 100 | 5.54 | 10.53 |
>
> Observe that the performance of MDP on BART and GPT2 is comparable to that on RoBERTa, indicating that it generalizes to other PLMs as well. We will add more thorough evaluation in the revision.
>
> Similar to prior work (e.g., RAP and ONION), this work focuses on defending against backdoor attacks in the context of text classification. As the backdoor attacks in other tasks (e.g., text generation) assume different threat models and attack/defense metrics, we consider extending MDP to such settings as our ongoing work.
>
> [1] Lewis, Mike, et al. "Bart: Denoising sequence-to-sequence pre-training for natural language generation, translation, and comprehension." arXiv preprint (2019).
>
> > Threat Model: The authors' assumption that the attacker introduces the backdoor directly into the PLM is not the only possible threat model. Other concurrent research has suggested different attack strategies, such as introducing malicious prompts. Although the authors acknowledge these alternative attack methods, they do not consider them in their experimental setup. Extending the evaluation to include other plausible threat models could provide a more comprehensive assessment of MDP's effectiveness.
>
> Following the suggestion, we have also explored an alternative threat model, in which the PLM is clean but the prompt introduces the backdoor. To generate the malicious prompt, we poison half of the few-shot samples from the non-target class with the trigger (as defined by BadNets and AddSent) and perform prompt tuning. The table below shows MDP’s performance under this alternative threat model on the SST-2 dataset.
>
> | Trigger | CA (%) | ASR (%) | FRR (%)  | FAR (%)  |
> | :--- | :--- | :--- | :--- | :--- |
> | BadNets | 70.18 | 61.47 | 10.17 | 33.26 |
> | AddSent | 86.01 | 49.07 | 5.88 | 57.26 |
>
> Notably, compared with attacks through poisoned PLMs, backdoor attacks via malicious prompts are less effective, as reflected in the degraded clean accuracy (CA) and attack success rate (ASR); meanwhile, MDP also seems less effective, mainly due to that the representations of clean and poisoned samples are less distinguishable on models with lower accuracy. We will evaluate MDP under other threat models (e.g., ones that combine poisoned PLMs and malicious prompts) in the revision.
>
> > Computational Overhead: The MDP mechanism involves comparing the representations of clean and poisoned samples under varying masking, which could be computationally expensive. The authors do not address this potential issue in the paper. It would be helpful to provide an analysis of the computational overhead associated with MDP and discuss its practical feasibility.
>
> Following the reviewer's suggestion, we have further evaluated the computation costs (in seconds) of MDP during both training and detection stages. The experiments are run on a workstation with a Nvidia RTX A6000 GPU.
> | Dataset  | Training (per epoch) | Detection (per sample) |
> | :--- | :--- | :--- |
> | SST-2 | 54s | 0.49s |
> | MR | 123s | 0.58s |
> | CR | 95s | 0.58s |
> | SUBJ | 38s | 0.58s |
> | TREC | 48s | 0.53s |
>
> Observe that both the training and detection in MDP are highly efficient under the few-shot setting, suggesting its practical applicability.
>
> > Fewer-Shot Scenarios: While the authors evaluate MDP under limited few-shot data (K as low as 4), real-world scenarios may involve even fewer data points, such as in one-shot or zero-shot learning scenarios. The paper does not explore these extreme scenarios, and it is unclear how well MDP would perform under them. Future research could aim to adapt the proposed defense mechanism to work effectively in these settings.
>
> In response to the reviewer's suggestion, we have evaluated evaluate MDP under extreme settings ($K$ = 1,2). The results on the SST-2 dataset with BadNets and AddSent attacks as reference attacks are shown below.
>
> | Attack | K | FRR (%)  | FAR (%)  |
> | :--- | :--- | :--- | :--- |
> | BadNets | 1 | 6.98 | 95.21 |
> | BadNets | 2 | 4.55 | 72.59 |
> | AddSent | 1 | 7.81 | 96.97 |
> | AddSent | 2 | 15.72 | 80.33 |
>
> Observe that MDP under $K$ = 1, 2 shots has large FARs, which can be explained as follows: MDP relies on the few-shot data as representational anchors; with very few anchors, MDP is not able to effectively distinguish clean and poisoned samples. However, note that such extreme settings also impede effective prompt tuning. For instance, prior work on prompt tuning (e.g., DART [1]) shows significant performance degradation with $K \leq 8$. We consider extending MDP to the fewer-shot settings as our ongoing research.
>
> [1] Zhang, Ningyu, et al. "Differentiable prompt makes pre-trained language models better few-shot learners." ICLR 2022.
>
> Again, we thank the reviewer for the valuable feedback. Please let us know if there are any other questions or suggestions.
>
> Best,
>
> Authors

---

> ### Author Response · Authors · 2023-08-18
>
> Dear Reviewer,
>
> Thank you again for your insightful feedback on improving this paper. We kindly seek confirmation on whether our response has adequately addressed your concerns. Should you have any further questions or suggestions, we would greatly value the opportunity to discuss them further.
>
> Best,
>
> Authors

---

> > ### Comment · Reviewer_fW9g · 2023-08-19
> > **Follow-up Questions**
> >
> > Thanks for the response. "To generate the malicious prompt, we poison half of the few-shot samples from the non-target class with the trigger (as defined by BadNets and AddSent) and perform prompt tuning. "  BadNets and AddSent are not prompt-based backdoors. It is not clear why the authors do not consider exploring existing prompt-based backdoors, like BadPrompt[1].
> >
> > [1]Cai, Xiangrui, et al. "Badprompt: Backdoor attacks on continuous prompts." Advances in Neural Information Processing Systems 35 (2022): 37068-37080.

---

> > > ### Author Response · Authors · 2023-08-19
> > > **Thank you for following up**
> > >
> > > Dear Reviewer,
> > >
> > > Thank you for the valuable feedback and follow-up!
> > >
> > > We would like to clarify that in this context, "BadNets" and "AddSent" indicate the trigger types, not specific attacks. To elaborate, "BadNets" uses rare words like "cf" as triggers, whereas "AddSent" uses neutral sentences such as "I watch this 3D movie." Prompt-based backdoor attacks generally involve two phases: trigger selecting and prompt tuning. The prompt-based attacks evaluated in our previous response differ from BadPrompt solely in the trigger selection phase. Our choice to use rare words and neutral sentences as triggers is mainly to align with other PLM-based attacks discussed in our paper.
> > >
> > > BadPrompt differs in its trigger selection. Instead of using rare words or sentences, it considers token combinations from samples in the target class as the trigger candidates; further, it selects the most effective trigger for each sample adaptively. In response to your suggestion, we also evaluate MDP against BadPrompt under the same setting as other prompt-based attacks ($K$ = 16 samples per class and 8 poisoning samples in prompt tuning), with results summarized below.
> > >
> > > | Trigger Type | CA (%) | ASR (%) | FRR (%)  | FAR (%)  |
> > > | :--- | :--- | :--- | :--- | :--- |
> > > | Rare Words | 70.18 | 61.47 | 10.17 | 33.26 |
> > > | Neutral Sentence  | 86.01 | 49.07 | 5.88 | 57.26 |
> > > | BadPrompt |  87.51  |  80.95 |  11.76  | 18.64 |
> > >
> > > Notably, compared with other trigger types, the adaptive triggers in BadPrompt lead to higher clean accuracy (CA) and attack success rate (ASR). However, since the prediction of poisoned samples hinges on these trigger tokens, they still exhibit high sensitivity to random masking, making them easily detectable by MDP.
> > >
> > > Overall, compared with PLM-based attacks, prompt-based attacks are less effective, as reflected in their lower CA and ASR. Meanwhile, MDP also seems less effective, mainly due to the difficulty in distinguishing "unsuccessful" clean and poisoned samples (i.e., a clean sample that is not classified into its actual class, and a poisoned sample that is not classified into the attacker's target class). For instance, if we filter out unsuccessful samples from consideration, MDP shows exceptional performance as below.
> > >
> > > | Trigger Type | CA (%) | ASR (%) | FRR (%)  | FAR (%)  |
> > > | :--- | :--- | :--- | :--- | :--- |
> > > | BadPrompt |  87.51  |  80.95 |  0.66  | 1.32 |
> > >
> > > Please let us know if our response has adequately addressed your questions. We also value any further questions or suggestions you might have.
> > >
> > > Best,
> > >
> > > Authors

---

> ### Author Response · Authors · 2023-08-20
>
> Dear Reviewer,
>
> Thank you again for your valuable feedback! As the discussion window is closing, we kindly seek confirmation on whether our responses have addressed your questions. Should there be additional questions or suggestions, we greatly value the opportunity to discuss them further.
>
> Best,
>
> Authors

---

### Official Review · Reviewer_6q5A · 2023-07-06

**Soundness:** 3 good
**Presentation:** 3 good
**Contribution:** 3 good
**Rating:** 7
**Confidence:** 3

**Summary:**

This paper proposes MDP(Masking-differential prompting), to defense PLMs as few-shot learners against backdoor attacks. The paper empirically validates that MDP outperforms baseline defenses.


**Strengths:**

1.The paper is well-written and easy to follow.

2.Investigating backdoor attack/defense under few-shot scenarios is interesting.

3.The evaluation is comprehensive.


**Weaknesses:**

1.Since the MDP is based on the observation: if the tirgger is (partially) masked, the language modeling probability of a poisoned sample tends to vary greatly. A statistics table supports this observation would be better.



2.There are invisible backdoor attacks such as stylebkd[1] and synbkd[2]. Does MDP also have a good performance on these more advanced attacks? (Or is it make sense to apply these two attacks under PLMs with few-shot setting?)



[1] Qi, F., Chen, Y., Zhang, X., Li, M., Liu, Z. and Sun, M., 2021, November. Mind the Style of Text! Adversarial and Backdoor Attacks Based on Text Style Transfer. In Proceedings of the 2021 Conference on Empirical Methods in Natural Language Processing (pp. 4569-4580).


[2] Qi, F., Li, M., Chen, Y., Zhang, Z., Liu, Z., Wang, Y. and Sun, M., 2021, August. Hidden Killer: Invisible Textual Backdoor Attacks with Syntactic Trigger. In Proceedings of the 59th Annual Meeting of the Association for Computational Linguistics and the 11th International Joint Conference on Natural Language Processing (Volume 1: Long Papers) (pp. 443-453).


**Questions:**

1.Whether you can extend MDP to a detection method?

**Limitations:**

The authors adequately addressed the limitations.

---

> ### Author Rebuttal · Authors · 2023-08-10
>
> We thank the reviewer for the valuable feedback on improving this paper! Please find below our response to the reviewer’s questions.
>
> > Since the MDP is based on the observation: if the trigger is (partially) masked, the language modeling probability of a poisoned sample tends to vary greatly. A statistics table supporting this observation would be better.
>
> In response to the reviewer's suggestion, we have measured the variance in the language modeling probability of both clean and poisoned samples due to random masking (under the BadNets attack). Under the default setting specified in Table 5 in the appendix, we have measured the masking variance of all clean samples in the testing set as well as their corresponding poisoned counterparts. The mean and standard deviation of these measures are reported in the table below. Across most datasets, there are noticeable gaps between clean and poisoned samples. However, also note that in some cases, the masking variance measures of clean and poisoned samples show large standard deviation, making it challenging to separate them using such measures directly, which motivates the design of MDP to use the more robust metric in Eq. 6.
>
> | Dataset | Clean Samples | Poisoned Samples |
> | :--- | :--- | :--- |
> | SST-2 | 0.0423 ± 0.0471 | 0.1055 ± 0.0752 |
> | MR | 0.0464 ± 0.0649 | 0.1463 ± 0.0725 |
> | CR | 0.0317 ± 0.0548 | 0.1274 ± 0.0721 |
> | SUBJ | 0.0180 ± 0.0464 | 0.1899 ± 0.0631 |
> | TREC | 0.0284 ± 0.0467 | 0.1538 ± 0.0627 |
>
>
> > There are invisible backdoor attacks such as stylebkd [1] and synbkd [2]. Does MDP also have a good performance on these more advanced attacks? (Or does it make sense to apply these two attacks under PLMs with few-shot setting?)
>
> In response to the reviewer’s suggestion, we have conducted a preliminary evaluation of MDP against StyleBkd [1] on the SST-2 dataset, with the results reported below.
>
> | CA (%) | ASR (%) | FRR (%)  | FAR (%)  |
> | :--- | :--- | :--- | :--- |
> | 92.97 | 81.66 | 4.04 | 51.96 |
>
> Notably, compared with insertion-based backdoor attacks, invisible backdoor attacks such as StyleBkd are less effective (lower ASR) but more challenging for MDP to detect (higher FAR). This may be explained as follows: insertion triggers are often local features (e.g., words and sentences), while invisible triggers tend to be global features (e.g., patterns of lexical choice and syntactic structures). Intuitively, compared with local features, global features are less sensitive to the random masking used in MDP. We consider extending MDP to the setting of invisible backdoor attacks as our ongoing work. One promising direction is to redefine the random masking operation to include random syntactic-structure perturbation or random style perturbation.
>
> > Whether you can extend MDP to a detection method?
>
> Thanks for the question. We would like to clarify that MDP is essentially a detection-based defense method, which falls into the same category as other baselines (e.g., STRIP, ONION, and RAP). We will revise the corresponding statement to make it more clear in the revision.
>
> Again, we thank the reviewer for the valuable feedback. Please let us know if there are any other questions or suggestions.
>
> Best,
>
> Authors

---

> > ### Comment · Reviewer_6q5A · 2023-08-14
> >
> > Thanks for your response.

---

### Decision · Program_Chairs · 2023-09-21

**Decision:**

Accept (poster)

**Comment:**

This paper introduces a defense to mitigate the vulnerability of backdoor attacks on language models. The reviewers all agreed the paper is interesting and raises new types of attacks. The main concerns shared by the reviewers, though, is the general applicability to other types of models. The authors respond to this criticism well and I agree that the technique appears promising, and novel. Therefore I recommend acceptance.